# An Evaluation of Indoor Sex Workers’ Sexual Health Access in Metro Vancouver: Applying an Occupational Health & Safety Lens in the Context of Criminalization

**DOI:** 10.3390/ijerph20031857

**Published:** 2023-01-19

**Authors:** Jennie Pearson, Sylvia Machat, Jennifer McDermid, Shira M. Goldenberg, Andrea Krüsi

**Affiliations:** 1Centre for Gender and Sexual Health Equity, Vancouver, BC V6Z 2K5, Canada; 2Interdisciplinary Studies Graduate Program, University of British Columbia, Vancouver, BC V6T 1Z4, Canada; 3Division of Epidemiology and Biostatistics, School of Public Health, San Diego State University, San Diego, CA 92182, USA; 4Department of Medicine, University of British Columbia, Vancouver, BC V6T 1Z3, Canada

**Keywords:** sex work, sexual health, occupational health and safety, migrant sex work

## Abstract

The criminalization of sex work has been consistently shown to undermine workers’ Occupational Health and Safety (OHS), including sexual health. Drawing on the ‘Guide to OHS in the New Zealand Sex Industry’ (the Guide), we assessed barriers to sexual health best practices among indoor sex workers in Metro Vancouver, Canada, in the context of ongoing criminalization. Part of a longstanding community-based study, this analysis drew on 47 qualitative interviews (2017–2018) with indoor sex workers and third parties. Participants’ narratives were analyzed drawing on a social determinants of health framework and on the *Guide* with specific focus on sexual health. Our findings suggest that sex workers and third parties utilize many sexual health strategies, including use of Personal Protective Equipment (PPE) and peer-driven sexual health education. However, participant narratives demonstrate how structural factors such as criminalization, immigration, and stigma limit the accessibility of additional OHS best practices outlined in the *Guide* and beyond, including access to non-stigmatizing sexual health assessments, and distribution of diverse PPE by third parties. Our current study supports the need for full decriminalization of sex work, including im/migrant sex work, to allow for the uptake of OHS guidelines that support the wellbeing and autonomy of all sex workers.

## 1. Introduction

Occupational Health and Safety (OHS) is a broad term used to refer to “any issue, task or condition in a workplace that may impact on the health and wellbeing of the people who are working there” [1]. In the context of in-person sex work, OHS is helpful in understanding sex workers’ health, safety, and wellbeing in a holistic way and through a labor rights framework. Through an OHS lens, we may consider workplace accessibility, adequate work hours and breaks, repetitive stress injuries, mental health, and access to a variety of Personal Protective Equipment (PPE), as just a few examples of occupational concerns important to sex work [1]. The use of an OHS lens also provides an opportunity for a more nuanced assessment of sex workers’ sexual health concerns and realities. However, the ongoing criminalization of sex work in most settings globally continues to hinder sex workers’ OHS across work environments, and has limited the potential of collaborative, indoor sex work spaces to implement OHS best practices.

In Canada, sex work OHS—in practice and as a framework—is restricted by the criminalization of various aspects of the sex industry, including ‘end-demand’ legislation enacted in 2014. Similar to other end-demand models taken up in over 50 countries around the globe [2], Canada’s sex work laws criminalize the purchase of sexual services and conducting sex work in public places [3]. Additionally, Canada’s laws criminalize third party supports, which include venue management, advertising, security personnel, and phone handlers, but also collaborative work environments and sex workers who provide support to other workers. The criminalization of third parties is based on the assumption of third parties as exploitative figures, and the conflation of sex work and trafficking [4,5,6]. This legislation effectively replicates conditions that had been deemed unconstitutional by the Supreme Court of Canada in 2013, on the basis of violating sex workers’ protected rights to security of the person [7]. In a long overdue review of these laws, the House of Commons recommended in June 2022 that client criminalization and the criminalization of third parties be upheld [8]. Such recommendations upholding the criminalization of sex work conflict with the OHS needs of sex workers by maintaining the framing of third parties as criminals rather than employers or coworkers, and sex workers as victims rather than workers [9]. End-demand legislation has specific implications for collaborative or managed indoor workspaces, where owners, managers, staff, and sex working co-workers face criminalization as sex work third parties.

Collaborative indoor venues pose significant potential for establishing and implementing occupational health and safety best practices in sex work and may provide substantial opportunity for collective organizing and peer support [1]. Prior Canadian research has demonstrated the potential of indoor work spaces as ‘enabling environments’ for sex worker collectivization and improved health outcomes (e.g., improved condom access) [10,11,12,13]. Despite this, the continued criminalization of sex work third parties limits sexual health best practices in managed spaces, by discouraging condom visibility in the workplace, the provision of condoms by third parties, as well as the use of sexual health outreach services [14,15]. Additionally, the criminalization of sex work more broadly perpetuates occupational stigma and prejudicial treatment in healthcare settings [14,15,16,17,18,19]. Among im/migrant (Community-based organizations have proposed ‘im/migrant sex worker’ as a term inclusive of the diverse persons (regardless of immigration status) who were born in another country and now carry out sex work in Canada (SWAN Vancouver, 2015). Our study uses ‘im/migrant’ to include all possible forms of immigration status) sex workers, impacts of end-demand laws are exacerbated as sex work criminalization, im/migration policy, racism, xenophobia, and culturally-specific stigma intersect resulting in unique barriers to health access [15,16,17,18,19,20].

In 2003, made possible by the efforts of the New Zealand Prostitutes Collective, New Zealand (NZ) became the first federal jurisdiction to decriminalize sex work. In NZ, sex workers (citizens and permanent residents only), their clients and third party supports can fully participate in sex work without criminal penalties, and sex work operates under the same health and safety rules as any other industry. Research from NZ since 2003 has shown that decriminalization has significantly improved OHS conditions and increased access to workplace protections and human rights among sex workers to whom the “Prostitution Reform Act” applies. These positive effects on sex workers’ OHS have, however, not been extended to NZ sex workers who are not permanent residents or citizens [21,22,23]. Shortly after the Prostitution Reform Act was passed, the NZ Department of Labour published ‘A Guide to Occupational Health and Safety in the New Zealand Sex Industry’ (the Guide) [24]. The *Guide* is based on previous OHS best practices developed by the Scarlet Alliance, an Australian collective of sex work organizations, and the Australian Federation of AIDS Organizations, with locally relevant input from the NZ Prostitutes Collective. Supplementary to regulations dictated by the Prostitution Reform Act and New Zealand’s ‘Health and Safety in Employment Act’, the Guide is meant to offer additional best practices to promote sex workers’ overall OHS. The Guide includes guidance specific to indoor venues, on topics such as sexual health (e.g., sexual health education; sexual health assessment); workplace amenities (e.g., cleanliness); psychosocial factors (e.g., security and safety from violence; complaints processes); and contains appendices relating to regulatory agencies, OHS reporting, and fact sheets on special OHS topics. As one of the only existing resources of its kind, the Guide may prove useful in settings beyond NZ for both sex workers and workplaces, and as a tool for evaluating sex workers’ access to best practices, even within criminalized contexts [25].

Since the introduction of the Prostitution Reform Act and the Guide, research in NZ has documented the positive impacts of OHS polices, as well as highlighted the gaps in OHS for im/migrant sex workers [26]. However, researchers have yet to utilize NZ’s best practices to evaluate other regulatory frameworks, including ‘end-demand’ criminalization, or experiences of sex workers in other geographical settings. Therefore, we drew on the Guide and a social determinants of health framework to assess indoor sex workers’ access to sexual health best practices under end-demand criminalization in Metro Vancouver. Given current gaps in literature related to OHS among sex workers in Canada, our objectives were to: (1) characterize participants’ OHS practices related to sexual health using the Guide as a framework, (2) assess the applicability and limitations of OHS guidelines within a criminalized setting, and (3) explore how intersecting structural determinants shape participants’ uptake of OHS practices under criminalization.

## 2. Methods

As part of a longstanding community-based study in Vancouver, this analysis drew on a total of 47 in-depth interviews collected between 2017–2018 with sex workers who work indoors, including managed, indoor venues and third parties who provide services for indoor sex workers (Table 1). This qualitative study is part of a longitudinal research project that evaluates how evolving approaches to sex work regulation shape sex workers’ health and safety, known as AESHA (An Evaluation of Sex Workers’ Health Access). The qualitative project runs alongside a longitudinal cohort of 900+ sex workers who work across diverse environments in Metro Vancouver. This research builds on community partnerships with sex work organizations since 2004 and has included experiential staff (current/former sex workers) on the project team since inception. The origins of AESHA are described in detail in previous publications [27].

For this analysis, sex workers and third parties (i.e., venue owners/managers/security/receptionists/phone handlers) working in massage parlours, beauty parlours, and private apartments (common indoor environments where sex work takes place in Metro Vancouver) were invited to participate in the context of ongoing AESHA outreach. Recruitment was facilitated by the AESHA outreach team who conduct weekly outreach to various indoor sex work venues and provide sexual health supports (i.e., deliver free condoms, lubricants and other harm reduction supplies, health counseling and referrals, and voluntary, confidential HIV/STI testing and treatment). Many study participants already had longstanding relationships with the project through regular contact with AESHA outreach workers. Eligibility criteria for qualitative interviews included currently working in an in-call venue (as a sex worker or third party) and being 18 or older. Outreach staff purposively invited participants reflecting diverse ages, lengths of time working in the sex industry, as well as third party roles. Additionally, we used snowball sampling within larger venues, where participants passed study information to their co-workers who were instructed to contact study staff if they were interested in participation.

Trained interviewers (including experiential staff) conducted semi-structured interviews in English/Mandarin/Cantonese with 47 participants between July 2017–November 2018 (Table 1). The interview guide explored four major topic areas: (1) criminalization and policing post-end demand law reform, (2) sex workers’ experiences with third parties, (3) access to health and social services, and (4) intersections between sex work and immigration. Interviews were conducted in a location selected by the participant (usually a private room in their workplace) and were 25 to 105 min long. Interviews were audio-recorded, translated into English when necessary, transcribed verbatim, and checked for transcription and translation accuracy. We maintained participant confidentiality by removing personal identifiers from all documents, and all participants provided informed consent and received $30 CAD for their time and expertise. The study received ethical approval from the Providence Health Care/University of British Columbia Research Ethics Board.

The research team discussed the interview content, emerging themes, and coding framework throughout data collection and analyses. Data analysis drew on deductive and inductive approaches [28], and drew on structural determinants [29] to explore the multilevel risk and protective factors shaping sex workers’ OHS experiences. We utilized a two-step approach to coding. The first and second author (JP, SM) coded the interview transcripts using a qualitative analysis software (NVivo), applying codes using a framework based on themes derived from the interview guide and participants’ accounts (e.g., immigration, experiences with third parties, peer training on sexual health), as well as key themes from the Guide that related to sexual health. For this analysis, we focused on sections of the Guide that offer best practices for sexual health education within indoor sex work environments, as well as suggestions around sexual health assessments and use and availability of PPE (e.g., condoms, lube), with a focus on the role of employers/venue managers, as well as sex work organizations and broader health services in supporting the uptake of OHS best practices. Examples of these codes included ‘Sexual Health Assessment for Sex Workers’ and ‘Personal Protective Equipment (PPE)’. All participant names presented in our results are pseudonyms to protect participants’ privacy. Furthermore, due to the small sample size in some categories, throughout the Results section, we removed identifying racial information in order to protect the anonymity of specific participants (Table 1).

## 3. Results

The results were drawn from a total of 47 in-depth interviews with sex workers and third parties working in indoor environments. There were 34 interviews with sex workers who work indoors, including managed, indoor venues; 25 interviews with third parties who provide services for indoor sex workers. Thirteen of the third parties currently held a dual role as a sex worker and third party. Thirty-nine participants worked in massage parlors or beauty parlors and eight worked in in-call (e.g., apartments), micro-brothels, or out-call spaces (work settings often less likely to be supported by third parties). Participants represented 18 separate sex work venues. Participants were aged 19–63 (median age: 39). In contrast to popular assumptions that position third parties as exploitative male ‘pimps’ [30,31], 17 of the 25 third parties in this study were also current/former sex workers and 22 identified as women. Consistent with the broader demographics of indoor sex workers in Metro Vancouver [4,6,32], 31 participants were im/migrants born outside of Canada, and more than half identified as Asian.

In the context of criminalization, participants’ narratives highlighted gaps in sexual health education, with a desire for increased access to education from public health educators, including in languages other than English. While sex workers received regular testing for sexually transmitted and blood-borne infections (STBBIs), few participants described being able to disclose sex work involvement to healthcare providers due to fear of stigma. These fears were amplified among sex workers who had previously experienced stigma in the healthcare system and those unable to access services in their first language. Multiple participants suggested that sex work-specific outreach services including sexual health testing and care would mitigate concerns around stigma and privacy. While many third parties provided basic PPE (i.e., condoms), others described the need to refuse condom distribution and discouraged visibility of condoms in the workplace over fears of criminal prosecution. Sex workers expressed needs for greater availability and variety of PPE to better suit their specific needs (e.g., allergies, sensitivities).

### 3.1. Sexual Health Education for Sex Workers, Their Clients, and Management

This section of the Guide includes information on: STBBIs; cleaning and disinfection of equipment; safer sex practices; and requirements for sexual health information to be shared in workplaces via diverse methods (e.g., print, verbal), and in different languages.

#### 3.1.1. Peer Education

The Guide emphasizes that employers (e.g., venue owners, managers) should be providing information to employees and to clients around safer sex best practices, and that such information should be prominently displayed in the workplace in language(s) with which the staff are familiar. However, when managed and collaborative sex work venues are criminalized, prominently displayed sexual health information is likely to be construed as criminal evidence. In contrast to the Guide, most participants in our current study indicated that rather than receiving sexual health education from third parties or having information displayed in the workplace, information-sharing was largely informal and peer-driven, with sex workers most frequently identifying coworkers as sexual health educators.

*“[A co-worker] mentored me, and then I had a really good sense of how to respect myself and keep myself fully [safe] so I had no worries and also respecting the people I was sharing, you know, time with. And also educating [clients], on things like that because in everyday life, even people’s personal lives that aren’t in the business, they might not know.”*—Participant 21, white cisgender woman, massage parlour worker and former third party.

*“We were like sisters. I learnt so much about the female body, and, the experiences I shared with those women [co-workers], and stuff they shared with me, I’ll keep for life.”*—Participant 14, BIPOC cisgender woman, worker and former third party.

Participants’ narratives suggest that sex workers learned information from other workers and continued to share that knowledge with co-workers, as well as clients.

#### 3.1.2. Gaps in External Resources

This section of the Guide also outlines that in the context of decriminalized work environments, local sex worker organizations, sexual health services, and other relevant health services should be available to sex workers for the purpose of sexual health education. In our study, many sex workers and third parties, particularly im/migrants, identified significant gaps in sexual health education, expressing desire to receive sexual health education from public health educators.

*“There’s so much education [targeting] the general public about [sexual health […] We [sex workers] and clients, need to take precedence in this. Like priority. Because we are providing these services.”*—Participant 13, white cisgender woman, manager/owner, and former worker.

In addition to wanting increased options for sexual health education for workers, multiple sex workers expressed frustration at the lack of sexual health promotion and education directed at clients. Participants commented on the need for clients to receive education on sex workers’ rights and occupational health, including the importance of STBBI testing and good hygiene. Further, no participants discussed the availability of sexual health education or information available in any language other than English.

When discussing sexual health education, participants identified ongoing gaps for sex workers, largely relating to lack of education in the workplace, lack of information in languages other than English, and gaps in education for clients. Opportunities exist to enhance sex workers’ occupational health via public health campaigns tailored for sex work venues, and that include education for clients.

### 3.2. Sexual Health Assessment for Sex Workers

The *Guide* defines ‘sexual health assessment’ as attending a sexual health service, family planning clinic, or general practitioner for sexual health assessment, counselling, and appropriate individual education. While the *Guide* suggests assessments take place twice a year and immediate response (10–14 days) following condom slippage or breakage, these health assessments should be at the discretion of individual sex workers and their trusted healthcare provider, and must be voluntary. Additionally, employers cannot demand disclosure of assessment results. The *Guide*, however, does not address the role of healthcare providers, clinics, or broader health systems, nor the role of continued sex work stigma, in shaping sex workers’ access to the recommended health assessments.

#### 3.2.1. Ongoing Stigma in Healthcare and Sex Workers’ Strategies of Resistance

In our current study, participants describe continued occupational barriers to sexual health assessments, which are shaped by criminalization and stigma. Almost all sex workers in our study reported accessing recent and regular STBBI testing; however, many participants described fear of disclosing their sex work to healthcare providers as an ongoing barrier, citing concerns about stigmatization and judgment.

*“At the family doctor’s office, you might feel embarrassed. They might say oh why are you doing this [STI test], so you feel scared.”*—Participant 4, East Asian cisgender woman, massage parlour worker.

*“I’m not having that conversation [about sex work] with [my male doctor]. No. … And I know as soon as I do, I will regret it. I know I will. He will change. I guarantee he’ll change. So no, I won’t. Maybe if it was a woman. I don’t know.”*—Participant 26, East Asian cisgender woman, massage parlour worker & third party.

Participants’ narratives highlight perceptions of sex work related stigma, which for some were also shaped by gendered power relations. Concerns regarding sex work- stigma stemmed from internalized stigma, or external stigma enacted through prejudicial judgement by healthcare providers. Several participants detailed that rather than disclosing sex work involvement as a reason for seeking testing, they would describe themselves as promiscuous, cheating on their partner, having been cheated on by a partner, or in a new relationship. A few sex workers described negative past experiences with healthcare providers expressing judgement about sex work or sexuality more generally. Experiences of sex negativity (shaming of sexual history or “promiscuous” sexual behaviors) impacted sex workers’ willingness to disclose work status to healthcare professionals and access to necessary sexual healthcare. As well, some sex workers described ‘clinic-hopping’ (visiting different clinics rather than regularly vising one clinic), accessing drop-in clinics for testing in various locations to avoid repeat visits and judgement regarding frequency of testing at one location.

*“I’ve been to a few places like at, youth clinics or clinics. And it’s all confidential, you can talk with them, you can get tested. And they’re non-judgmental… [Y]ou tell them you’re a sex worker, they’ll do all the tests you need. Cause I feel like if you go to the doctor’s office, they’ll be like hey, why are you getting tested so often?”*—Participant 44, white cisgender woman, massage parlour worker & former third party.

Participants described many strategies that they used to navigate sex work disclosure to avoid experiences of stigma and discrimination when seeking sexual health assessments. However, the reliance on such strategies demonstrates sex workers’ inequitable access to safe and supportive health services.

#### 3.2.2. Experiences of Racism and Xenophobia When Accessing Sexual Health Assessment

Participants who identified as racialized [33] shared concerns about judgment from healthcare providers relating to intersections of sex work and race (i.e., as one participant described, “judgments about ‘brown girls’ in sex work”) and Asian participants frequently brought up culturally specific stigma around STI testing.

*“They don’t dare go to see the doctor when they are sick or maybe are infected with something. It really is like this! So they bring their own medication from China to self-medicate. They say they feel ashamed to go see the doctor. Even though they are legal, the fear is still there. They are afraid other people will find out about their work. They feel it is a very shameful work. Other people will look at them differently.”*—Participant 27, East Asian cisgender woman, third party.

Adding to reduced options for health services due to stigma, some sex workers faced the additional challenge of finding a healthcare provider fluent in their first language.

*“Mostly just don’t know where to look… If there is Chinese [spoken], then I’m ok.”*—Participant 18, East Asian cisgender woman, massage parlour worker & third party.

#### 3.2.3. Peer-Led Alternatives for Sexual Health Services

The Guide generally describes sexual health assessment as taking place within the broader health care system (public clinics or family doctor offices) and does not indicate the role of sex work organizations or discuss peer models as an avenue for sexual health assessment. However, in our current study, several participants expressed enthusiasm for sex worker-led outreach services in indoor sex work spaces. Reasons included: ease of testing during work hours in the work environment; increased privacy (e.g., compared to seeing a family doctor) and availability, particularly while parenting; reduced stress related to non-stigmatizing healthcare providers; and increased self-esteem related to perceived government endorsement of sex workers’ health access. In contrast, fewer sex workers preferred to access testing through their family doctor or another healthcare professional at a fixed location. The primary reason given for preferring testing at a fixed location was privacy from co-workers, with one worker noting that in her workplace ‘*walls are thin*’ and sex workers may wish to discuss additional health issues while getting sexual health testing (e.g., mental health) in a confidential setting.

Most sex workers in our study described accessing regular testing. However, due to the criminalized environment and intersecting stigmas many faced, barriers to sexual health assessments, including voluntary, convenient, and language-specific testing, were common. Participants identified gaps in sexual health assessment access, including lack of services in languages other than English as well as opportunities for non-stigmatizing sex work-specific health assessment programs to be made available via outreach and fixed-point locations to accommodate diverse concerns around privacy.

### 3.3. Personal Protective Equipment (PPE)

As outlined by the *Guide*, PPE for sex workers and clients includes condoms, water-based lubricants, latex and non-latex gloves, and materials required to maintain equipment and facilities (e.g., disinfectant). The *Guide* requires PPE to always be accessible during work hours; requires provision of PPE variety (e.g., various sizes and materials); that PPE be stored in a cool, dark area; and describes guidelines on PPE disposal, tracking PPE use with multiple partners by using a colour-coded condom system, cleaning standards, and PPE training requirements.

#### 3.3.1. Access to PPE

Despite intense stigma around sex work and sexual health, narratives in our current study describe sex workers as having a strong knowledge of best practices and as major advocates of PPE and safer sex. In our study, when asked about sexual health or PPE, most participants discussed condoms and lubricants, and many stressed the importance of proper use of PPE and the availability of effective supplies.

*“I make sure I check every fuckin condom.”*—Participant 42, BIPOC cisgender woman, massage parlour worker.

In our study, cleaning supplies including disinfectants were typically provided by sex workers, with a few situations where third parties provided such supplies. Sex workers described PPE disposal, tracking, and training as provided mainly by other sex workers and third parties with previous sex work experience. Participants accessed condoms through various methods, including retail outlets, public health locations (e.g., clinics), outreach workers, and third parties. The *Guide* outlines that venues and employers should provide all needed PPE, as part of the employer’s general duty to ensure employees’ safety. Therefore, employees should not be required to pay for anything required to meet an employer’s legislative duties. However, in our current study, where sex work third parties are considered criminals rather than employers, not all third parties supplied condoms for sex workers. Some participants described third parties refusing condoms distributed by outreach workers and generally discouraging visibility of condoms in the workplace due to fear of criminal prosecution.

*“Each of us had [condoms] with us but we did not leave them in the parlour… Because the owner did not allow in case of raid.”*—Participant 13, white cisgender woman, massage parlour third party & former worker.

As outlined by the above quote, in some workplaces third parties’ awareness of their criminalized role resulted in reduced access to condoms. However, some sex workers specifically noted being supplied condoms at work as a benefit of working in that type of workplace or for that particular manager. Despite fears of criminalization, third parties who described supplying condoms for sex workers expressed concern for sex workers’ health and wellbeing, and some sex workers and third parties described situations in which managers educated or reminded sex workers about condom use.

Sex workers whose third parties provided PPE described condom and lubricant availability as adequate but not ideal in variety due to allergies or ingredient preferences, and lacking consistency of provision between and within workplaces. One participant described purchasing their own PPE, not due to a total lack of supplies, but more so a lack of supply variety provided by third parties.

*“Like I’m very, picky about certain things cause certain condoms break, certain condoms don’t break. Uh certain condoms will give you infections too. Depending on your body. So, I don’t like when owners also tell me what you do or don’t have to do in the room.”*—Participant 14, BIPOC cisgender woman, massage parlour worker.

#### 3.3.2. PPE Usage

In addition to the importance of sufficient access to a variety of PPE supplies and condoms, some sex workers shared concerns around, as well as strategies to avoid, condom slippage and breakage.

*“After oral, we take off the condom, wash our hands, put on a new one for full service because lipstick or gloss can break down your condom.”*—Participant 13, white cisgender woman, massage parlour third party & former worker.

The *Guide* includes specific guidelines for events of condom slippage and breakage, explicitly stating a worker’s right not to provide a service to a client who refuses to use a condom or who attempts to break or remove a condom during service. In our study, sex workers reported upsetting incidents of accidental, and intentional condom breakage by a client, with no clear procedures for responding to such events at work. In the absence of OHS guidelines, participants’ narratives indicated that these incidents were best supported by coworkers and third parties providing emotional comfort and advice, as described by one participant:

*“[I]f people have condoms that break um, and then they start crying, people are there to hold you and to hug you and to, to touch you and just to say you’re okay, like it happens.”*—Participant 38, white cisgender woman, massage parlour worker.

With regards to guidelines on condom breakage or slippage, the *Guide* also describes that “Employers should not only require condom use but should also identify condom use and other safer sex practices clearly to employees and clients as the standard, expected practice of the establishment”. Our current study found that under end-demand criminalization, PPE policies are more likely developed by individual workplaces, managers, or each individual worker, but sex workers remain vulnerable to coercion or restrictions due to power imbalances and impacts of criminalization. In our study, multiple sex workers and third parties expressed frustration at the inconsistency in condom use policies across the local indoor sex industry. Condom policies across venues can be based on competition within the industry, opportunity for higher rates, and pressure to provide unprotected services from clients or third parties. Some sex workers, particularly racialized sex workers, described using workplace condom policies as a deciding factor in where to work, declining employment at venues with no-condom policies. Sex workers noted inconsistency in condom use between types of workplaces and between individual workplaces, with some sex workers switching workplaces to manage risk.

Overall participants described regular access to PPE. However, in contrast to the *Guide*, the purchasing and distribution of PPE was largely the responsibility of workers. Participants specifically noted criminalization of third parties as a barrier to ensuring PPE provision for sex workers, as PPE in the workplace could be used as evidence of criminal activity. Opportunities exist to ensure third parties are able to provide PPE to sex workers, including a variety of condom types (e.g., materials, fits), lubricants, and latex and non-latex gloves. In addition, participants’ experiences with no-condom policies suggest the need for an improved policy that allows for enabling environments that are supportive of workers’ autonomy.

## 4. Discussion

In decriminalized environments, the *Guide* describes OHS best practices for indoor sex workers as including bi-annual sexual health assessments, sexual health education within the workplace, consistent access to diverse, appropriate, and effective PPE provided by employers, as well as condom mandates. Our study findings suggest that within a suboptimal context of criminalization, indoor sex workers utilize many strategies to support sexual health, including peer-driven sexual health education and PPE usage, as relevant to their own work and personal preferences; however, gaps remain. Under the constraints of third party criminalization, third parties attempt to support workers’ OHS, but must navigate the risks and benefits of having sufficient condoms in the work place, offering sexual health education, or collaborating with sex work organizations and outreach services. While decriminalization is essential in supporting sex workers’ OHS, as found in NZ, current findings also demonstrate the need to address intersecting structural inequities in addition to decriminalization. Our findings demonstrate several structural factors shaping sex workers’ OHS external to their work environment. Participants in our study, similar to other contexts [19,34,35], described ongoing barriers to structural OHS supports such as the need for non-stigmatizing and culturally safe sexual health services and testing, and tailored sexual health education for workers and clients. For racialized and im/migrant sex workers and third parties, OHS practices and the ability to support OHS were also shaped by racism, xenophobia, and im/migration policy [15,20,26,32], as well as the scarcity of sexual health resources available in multiple languages.

While the Guide assumes equal access to OHS best practices for all sex workers across work environments in NZ, sex work in NZ remains criminalized for im/migrants, limiting access to OHS protections for this population [21]. Research from NZ finds that the “two-tiered” approach to decriminalization has restricted applicability of the *Guide* and thus better working conditions to citizens and residents. Whereas im/migrant sex workers, particularly those who do not speak English and who are working in informal brothels, have reduced power to negotiate and enact OHS best practices, and are most vulnerable to exploitative workplace practices [26]. Similarly, the findings of our study indicate that racialized and im/migrant sex workers face compounding barriers to OHS, as im/migrant sex work is criminalized, as well as their third party supports. Sex workers in our study experiencing intersecting forms of oppression, such as precarious im/migration status and/or language barriers described obstacles to health assessments and PPE, some of which were mitigated by supportive third parties, as well as gaps in sexual health education and health assessment services offered in languages other than English. The unique realities of im/migrant sex workers, as described in our study and previous research, underscore how intersecting social locations and structural factors determine the relevancy and effectiveness of sex work OHS guidelines [17,22,26]. Therefore, it is essential that an evaluation of sex workers’ uptake of best practices, as described in the Guide or otherwise, considers the compounding impacts of im/migration policy and ongoing criminalization.

The Guide explicitly addresses the availability of PPE and indicates that workers should not bear any financial burden for acquiring appropriate equipment when working in managed venues. In our setting, where third party supports remain criminalized, participants described hesitancy of their managers to provide PPE such as condoms, but overall, participants described high levels of access and uptake of PPE through various distribution methods, and knowledge of PPE use best practices despite formal policy or training. These findings build on prior research indicating that collaborative work environments and access to third party supports can promote an increased uptake of condoms and sex worker-led outreach [11,13,14,16]. Utilizing the Guide was also helpful in framing sex workers’ PPE needs beyond ‘access to condoms vs. no access to condoms’. Participants’ discussions of PPE extended to the need for a variety of products and materials based on allergies or preferences. Participants narratives also suggest ongoing discrepancies in condom use policies across workplaces. When condoms continue to be used as criminal evidence, and indoor venues are subject to police inspection, it is more likely for third parties to limit condom visibility and distribution, hindering sex workers’ ability to consider best practices and utilize PPE in ways that feels best for themselves [16]. In decriminalized settings, the Guide indicates that condoms should be *required* by employers; however, critical attention is needed on condom mandates and punitive approaches. Global research has identified greater *access* to PPE (via decriminalization, uptake in outreach or sanctioned work environments), as a key intervention for improved OHS among sex workers [14,34,36]. However, policies and regulations that are overly prescriptive in dictating sexual health practices, including PPE usage [36] and mandatory testing [37,38,39], have often been applied in coercive ways and can impede sex workers’ health and human rights. Future guidelines of PPE use must be done through thoughtful consultation with sex workers while supporting sex workers’ individual autonomy. In supporting sex workers’ autonomy regarding PPE use, it may be most helpful if third parties create an enabling environment for PPE use and take up the Guide’s recommendation to “support an employee’s right not to provide service to a client who refuses to use a condom”. However, in order to create such an enabling environment, we must end the criminalization and policing of third parties and managed venues.

Similarly to the ways in which third party criminalization limits the distribution of appropriate PPE, participants described gaps in sexual health education, within the workplace and more broadly. In contrast to the Guide and echoing prior Canadian research [18], participants in our current study indicated that rather than receiving formal or print-based sexual health education within the workplace, information-sharing was largely verbally-facilitated, informal, and peer-driven. However, current findings and prior qualitative research suggest that third parties with prior sex work experience play a crucial role in providing sexual health education yet they remain criminalized [6,16]. The Guide also outlines that sex work organizations and public health agencies should take on sexual health education efforts. Our findings and existing research suggest that sex workers have a keen interest in sexual health campaigns tailored for sex workers as well as clients [40,41]. Participants in this study expressed frustration that educational efforts are generally focused on sex workers who are already largely aware of OHS best practices, rather than clients. Recommendations listed in the Guide, including sexual health information available in work places and tailored to workers and clients, would be helpful in increasing knowledge among clients and reducing workers’ burden of being the sole sexual health experts within the industry.

Extensive research from Canada and around the globe has captured the ways that sex work stigma is exacerbated by laws that criminalize sex workers, third parties or any facet of the industry [17,35,42,43,44]. Results from our current study describe the impacts of stigma on sex workers’ access to sexual heath assessments, as well their experiences within various healthcare settings. Participant’s narratives highlight fears associated with disclosure of sex work involvement, exacerbated by experiences of misogyny, racism and xenophobia, and sex negativity. In order to reduce the harm caused by healthcare providers, sex workers may avoid regular sexual health assessments or “clinic hop”. These findings resonate with past research examining health access in the context of end-demand laws and im/migration policies that restrict sex work involvement [15,17,45]. In the Guide, it is recommended that sex workers access regular sexual health assessments as well as immediate assessment in the event of condom slippage or breakage. The Guide is implicit in its assumption that in the context of decriminalization, sex workers will have access to safe and stigma-free health services, and that regular and emergency sexual health assessments are a feasible best practice. However, what remains beyond the scope of the *Guide*, is the role of occupational stigma in shaping sex workers’ access to and experiences with sexual health services. Recent research from NZ has found that even decades after decriminalization, the sex work stigma persists [46], and many workers still experience concerns around disclosure and such concerns are exacerbated based on other intersecting facets of workers’ identities, including race [47]. Findings from our current study, as well as NZ, suggest the need for interventions beyond decriminalization to reduce the sex work stigma, including best practices directed at healthcare providers on stigma-free and culturally safe care for sex workers, and increased options for sex worker-led services.

### 4.1. Policy Implications

Based on our research and supported by prior literature on sex workers’ OHS, we make the following recommendations:The decriminalization of all aspects of sex work in Canada, including the decriminalization of third parties (e.g., managers, receptionists), allowing third parties to actively support sex workers’ OHS and sexual health education, while reducing policing of workspaces.The decriminalization of im/migrant sex work through the repeal of immigration policies which conflate migrant sex work with trafficking, to ensure the safety, health, and human rights of im/migrant sex workers.Improved supports for im/migrants to promote their health, wellbeing, and equal access to employment, income supports, and health services upon arrival to CanadaIncreased funding for multilingual, sex worker-led occupational health programs. Such programs should include both outreach and fixed-point components and should be available in languages other than English (e.g., Mandarin).The development of resources by and for indoor sex workers, similar to the *Guide*, that increase access to supports and promote sex workers’ OHS and autonomy.Further research on sex workers’ OHS needs beyond sexual health.

### 4.2. Strengths and Limitations

This study has several strengths and limitations. Challenges in reaching criminalized populations may have constrained participation, as im/migrant workers and third parties fearing surveillance may be more likely to have declined engagement with this research. However, the research team has worked to mitigate this through building longstanding relationships with sex work venues throughout a decade of outreach. Considerable overlap between sex worker and third party roles is a major strength of this study, as it offers dual perspectives on the impacts of third party criminalization in managed indoor settings. Due to the small sample sizes of Black and Latinx sex workers and third parties, we aimed to protect participant anonymity through use of a broader BIPOC descriptor. Therefore, this research is unable to describe the unique experiences of Black, Latinx, or other BIPOC indoor sex workers. Further research is needed, which engages with larger and more diverse groups of BIPOC sex workers, to address the impacts of diverse forms of racism, including anti-Black racism on indoor sex workers’ access to sexual health best practices. Lastly, we recognize that sex work OHS is much more broad than sexual health needs, and we encourage future research to address the various components of sex workers’ working conditions and wellbeing through the lens of OHS best practices.

## 5. Conclusions

Our study findings suggest indoor sex workers utilize many strategies to support their sexual health, including peer-driven sharing of sexual health education, and PPE usage, as relevant to their own work and personal preferences. Third parties, as well, aim to support workers’ OHS by providing PPE and informal sexual health education. However, such strategies are constrained by the risks of criminalization and ongoing occupational stigma. Participants describe continued barriers to stigma-free sexual health assessment, as well as formalized sexual health education, including services and resources provided in multiple languages, and diverse PPE provided by their workplace. The Guide describes many important OHS best practices relevant to indoor sex workers, including those pertaining to sexual health. When applied to the experiences of sex workers working within managed indoor venues, in the context of end-demand criminalization, the impacts of pervasive sex work stigma and punitive im/migration policy on limiting sex workers’ uptake of best practices become more apparent. Barriers to OHS are exacerbated by third party criminalization, occupational stigma, and intersecting oppressions such as racism, xenophobia, and im/migration policy. In addition to full decriminalization to allow for sex worker-led OHS guidelines, we must also address structural issues beyond criminalization, and ensure that guidelines are responsive to structural vulnerabilities including im/migration status and stigma. There is an urgent need for full decriminalization of sex work, including for im/migrant sex workers, in conjunction with anti-stigma and education efforts, to promote enabling work environments and the uptake of inclusive and effective OHS guidelines.

## Figures and Tables

**Table 1 ijerph-20-01857-t001:** Individual and structural characteristics of indoor sex workers and third parties in Metro Vancouver, Canada (*n* = 47), AESHA 2018.

Characteristic	*n* (%) **
Age (Mean age)	39 (19–63)
**Migration Status**	
Canadian-born	16 (34%)
Im/migrant	31 (66%)
**Racialization**	
BIPOC * (e.g., Black, Asian, Latinx)	13 (27%)
Black	4 (8%)
Latinx	5 (11%)
Asian (e.g., East Asian and South Asian)	21 (51%)
White	17 (36%)
**Gender Identity**	
Women	42 (89%)
Men	3 (6%)
Non-binary or gender minority	2 (4%)
**Sex work involvement**	
Sex work experience, no third party role	21 (45%)
Dual role (third party and sex worker)	13 (28%)
Third party, past sex work experience	4 (8%)
**Type of workplace**	
Massage parlour	37 (79%)
Brothel and micro-brothel (>2 workers)	3 (6%)
In-call (i.e., apartment)	2 (4%)
Out-call/hotel	3 (6%)
Beauty spa	2 (4%)

* Black/Indigenous/People of Colour (Due to the small sample size in some categories, throughout the results section we removed identifying racial information in order to protect the anonymity of specific participants) ** Participants were able to select more than one option, resulting in sums of >100%.

## Data Availability

The data presented in this study are available on reasonable request from the corresponding author.

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
