# Peer review of "An Evaluation of Indoor Sex Workers’ Sexual Health Access in Metro Vancouver: Applying an Occupational Health & Safety Lens in the Context of Criminalization"

_ijerph, 2023, doi:10.3390/ijerph20031857_

Round 1

Reviewer 1 Report

Please see attached file. Thank you!

Author Response

Reviewer #1

Line: 25: remove the period after Equipment

Line 51: why is Globe capitalized?

Line 70: this sentence is confusing

Line 86: “…. to whom ….. applies.”

Thank you for the above suggestions. We have incorporated the above four comments.

Line 123: clarify is 2004 is when the organization or the research began

Thank you for highlighting this. We have specified the timeline for our data collection (2017-18) earlier in the paragraph.

Line 126: specify what you mean by “elsewhere”

Line 138: “… diverse ages and lengths of time working in the sex industry, as well as….”

Line 158: sex workers’ OHS experiences

Line 171: capitalize Results, and add a comma after “section”

Line 258: “…. languages with which the staff are familiar.”

Line 277: “…. co-workers, as well as clients.”

Line 281: remove the f from study

Thank you for the above suggestions and for catching some of our errors. We have made the above seven corrections.

Line 290-299: please make is clearer that the first paragraph is about the clients and the second is about the workers

Thank you for this suggestion. The second paragraph is meant to be a concluding paragraph for the theme of sexual health education. We have revised the paragraph to make this clearer.

Line 336: “ and to access….”

Line 337: “clinic hopping, which refers to….”

Line 367: do you mean that the Guide “describes” instead of “refers to?”

Line 393: “…. requires PPE to always be accessible….”

Line 403: possibly remove “In our study,”

Line 460: participants’ narratives

Line 518: I would avoid using contractions

Line 639: “Third parties, as well, ….”

Thank you for the above suggestions. We have made the above eight revisions.

Line 478: what do you mean by “racialized” sex workers?

Line 520: please explain what you mean by “racialized” sex workers earlier in the paper

Thank you for highlighting this opportunity to expand on our use of the term “racialized”.

Racialization is the very complex and contradictory process through which groups come to be designated as being part of a particular "race" and on that basis subjected to differential and/or unequal treatment. “racialization [is] the process of manufacturing and utilizing the notion of race in any capacity” (Dalal, 2002, p. 27).

The process by which people are identified by racial characteristics is a social and cultural process, as well as an individual one. That is, a social order might racialize a group through media coverage, political action, and the production of a general consensus in the public about that group (The Alberta Civil Liberties Research Centre).

At line 370, where “racialized” is first referenced, we have added a citation so readers may have access to a full definition if needed.  

Sometimes The Guide is italicized, sometimes it’s not (or “the” is, or “Guide is,” and so on).

Thank you for identifying this inconsistency. We have ensured consistent formatting of the Guide” throughout.

Author Response

Reviewer #2

  • The material is valuable due to the difficulty of accessing the research sample and the longitudinal aspect of the research.

Thank you for acknowledging the research as valuable, and for your thorough review and constructive feedback.

  • I suggest that thorough separation of the specific research problems that were analysed should be made in the Introduction.

Thank you for this suggestion. We have included a more specific description of our research objectives at the end of the introduction.

  • Furthermore, I also propose that the Authors emphasise (either in the Introduction or in the Discussion) that the consequences of sex work (both immediate and longterm), stem not only from contracting sexually transmitted diseases, the brutality of clients during intercourse, or the use of psychoactive substances, but also from accepting the role of an object for the clients' sexual satisfaction. As a result, service providers' self-esteem decreases significantly. Thus, they require, as part of OHS, emotional support and also support in terms of lifestyle changes (giving up commercial sexual activity).

We appreciate your suggestion, and the importance of discussing mental health and emotional supports in relation to sex work occupational health. As our paper is specific to sexual health outcomes, we believe this is beyond the scope of the current manuscript. We have mentioned in our recommendations and in the limitations the need for more diverse discussions of sex work OHS.

As well, in our labour-rights based framing of sex work, and by our understanding of sex work as not inherently harmful, we feel it would not be appropriate to describe sex workers’ as “accepting the role of an object”, nor to assume that sex workers experience poor self-esteem. These themes as well were not present in our study data. Regarding the need for “support in terms of lifestyle changes”, rather than narrowly emphasizing “existing strategies”, our framing instead promotes autonomy and self-determination. Our recommendations support need for better working conditions within sex work, as well as equal access to other industries and types of employment for those wishing to change industries.

  • In the Methods, I would recommend to describe the stages of self-analysis with more clarity. It would be advisable to structure the Results of the research in a transparent way, e.g. 3.1 Education...:
    Sources of knowledge on sexual health (existing and desired by respondents),
    • Extent of information received so far and declared educational needs of respondents,
    • Determinants of educational deficits as seen by respondents.

Thank you for this suggestion as to how we may improve the organization of the results section. We have added relevant subheadings throughout.

  • It is vital that all conclusions are accurately and precisely justified by the statements of the respondents (in the material presented, they are sometimes general, and sometimes cursorily argued; justifications can also be included in the Appendix). While the Results provide a detailed description of the guidelines contained in A Guide to Occupational Health and Safety in the New Zealand Sex Industry (the Guide), the self-referenced findings are quite broad, or refer to a small number of guidelines/recommendations from the manual. It would be worth extending it, should the Authors have the material.

We fully agree with the reviewer, that it is vital to clearly link our findings to participant responses. Here, we have listed all of the sexual health guidelines from the Guide, and how they are highlighted in the discussion section, relating to participant’s quotes:

STBBIs; cleaning and disinfection of equipment; safer sex practices; and requirements for sexual health information to be shared in workplaces via diverse methods (e.g., print, verbal), and in different languages:

“Sex workers in our study experiencing intersecting forms of oppression, such as precarious im/migration status and/or language barriers described…gaps in sexual health education and health assessment services offered in languages other than English” (Lines 571-75).

“Participants described gaps in sexual health education, within the workplace and more broadly. In contrast to the Guide and echoing prior Canadian research 18, participants in our current study indicated that rather than receiving formal or print-based sexual health education within the workplace, information-sharing was largely verbally-facilitated, informal, and peer-driven…Our findings and existing research suggest that sex workers have a keen interest in sexual health campaigns tailored for sex workers as well as clients 39,40. Participants in this study expressed frustration that educational efforts are generally focused on sex workers who are already largely aware of OHS best practices, rather than clients“ (Lines 624-30).

The Guide suggests sexual health assessments take place twice a year and immediate response (10-14 days) following condom slippage or breakage, these health assessments should be at the discretion of individual sex workers and their trusted healthcare provider, and must be voluntary.

“the findings of our study indicate that racialized and im/migrant sex workers face compounding barriers to OHS, as im/migrant sex work is criminalized, as well as their third party supports. Sex workers in our study experiencing intersecting forms of oppression, such as precarious im/migration status and/or language barriers described obstacles to health assessments and PPE, some of which were mitigated by supportive third parties, as well as gaps in sexual health education and health assessment services offered in languages other than English” (lines 569-75).

“Results from our current study describe the impacts of stigma on sex workers’ access to sexual heath assessments, as well their experiences within various healthcare settings. Participant’s narratives highlight fears associated with disclosure of sex work involvement, exacerbated by experiences of misogyny, racism and xenophobia and sex negativity. In order to reduce harms caused by healthcare providers, sex workers may avoid regular sexual health assessments or “clinic hop” “(Lines 633-38)

As outlined by the Guide, PPE for sex workers and clients includes condoms, water-based lubricants, latex and non-latex gloves, and materials required to maintain equipment and facilities (e.g., disinfectant). The Guide requires PPE to always be accessible during work hours; requires provision of PPE variety (e.g., various sizes and materials); policy in the case of condom breakage or slippage and recommends a condom mandate within workplaces.

“In our setting, where third party supports remain criminalized, participants described hesitancy of their managers to provide PPE such as condoms, but overall, participants described high levels of access and uptake of PPE through various distribution methods, and knowledge of PPE use best practices despite formal policy or training.” (Lines 583-86).

“Participants’ discussions of PPE extended to the need for variety of products and materials based on allergies or preferences. Participants narratives also suggest ongoing discrepancies in condom use policies across workplaces” (589-96).

  • In addition, it would be interesting to show how many of the respondents reported that they had certain problems, e.g. had fears of stigma from health professionals, did not receive condoms for free, etc. (e.g. 20 people reported that they did not tell their doctor about having sexual services due to the fear of being stigmatised). This would highlight the magnitude of the problems faced by sex workers.

Thank you for this suggestion. Given this is a qualitative study focused on the lived-experiences of participants rather than on quantifying results it is not our priority to quantify experiences of participants. Rather, the strength and focus of qualitative research lies in providing ‘rich’ descriptions of the lived experiences of participants as outlined in our manuscript. Therefore, as customary with qualitative research, we refrain from quantifying our results as this does not carry much meaning in the context of our methodological approach including sample size and approach to sampling. That said, we have included quantifiers such as “most; many; some; or few” throughout the results section.

  • The Authors' demand for the full decriminalization of sex work is right. However, it should be noted in the Discussion that not only is this necessary for health and safety reasons (e.g. access to condoms, medical care), but it is equally significant that decriminalisation would increase the chances of getting help to change one's lifestyle and thus satisfy one's material needs without the necessity of providing paid sex work.

Thank you for this suggestion. Our framing, of drawing on the NZ Guide as an analytical lens,  certainly supports sex workers equal access to opportunity and autonomy over working conditions. However, we feel that arguing for a “lifestyle change” among sex workers is beyond the scope of the results of this study. That said, our findings did suggest that many im/migrant sex workers faced significant barriers to other types of employment upon arrival to Canada, we have recommended “Improved supports for im/migrants to promote their health, wellbeing and equal access to employment, income supports, and health services upon arrival”.

  • As regards the call for the full decriminalization of all third parties, I have some doubts (due to the risk of human trafficking). Please, in order to clarify any uncertainty, expand on the description of your concept of decriminalization of third parties.

Thank you for this comment and for the opportunity to further clarify the importance of third party decriminalization. Contrary to stereotypes and the conflation of third parties and “the exploitative male “pimp” figure, ample research from Canada and around the globe has found that third party roles are often occupied by current and former sex workers and a majority of third parties are women, which is also true in the case of our current study.  Sex work third parties are diverse, like in other service industries. As described in our results and elsewhere, third parties in sex work provide client screening, security and sexual health resources for sex workers which promote sex workers’ occupational health and safety. However, under Canada’s current legal model, there is no distinction between supportive third parties (bosses and coworkers), or exploitive figures. This ongoing criminalization and policing of sex workers and their workspaces, through third party criminalization, reproduces the same harms as previous laws. Therefore, to reduce the impacts of policing, we must decriminalize all aspects of sex work.

We, as well, understand the seriousness of human trafficking that exists within many industries, however, it is important that policy does not conflate trafficking with sex work, and that sex work is not inaccurately exceptionalized as a main site of trafficking.

We have updated our introduction and discussion section with similar text to above, and have included additional citations to support our calls for third party decriminalization.
